# A Parallel Resonant Converter Polynomial Model Implemented in a Digital Signal Controller

Evode Rwamurangwa * , Juan Díaz González , Pedro José Villegas Saiz , Juan Antonio Martín-Ramos and Alberto Martin Pernía

Department of Electric, Electronic, Computer and Systems Engineering (DIEECS), University of Oviedo, 33203 Gijon, Spain; jdiazg@uniovi.es (J.D.G.); pedroj@uniovi.es (P.J.V.S.); jamartin@uniovi.es (J.A.M.-R.); amartinp@uniovi.es (A.M.P.)

* Correspondence: uo241006@uniovi.es or ing.rw.evode@gmail.com; Tel.: +25-078-883-8816

**Abstract:** Due to their exceptional performance in coping with large variations in output voltage and current, parallel resonant converters (PRC) are commonly used in high-voltage applications. The incorporation of step-up transformer parasitic components as part of a power topology, on the right duty and a suitable switching frequency, determines the high efficiency and wide variety of applications with PRC. Switching losses are reduced in the same topology by tracking and running on the optimum mode for each power and voltage by a set frequency and duty. The PRC's static model behaviors, under optimum operating circumstances, are illustrated. The equivalent polynomial model is used to quickly compute the switching frequency and duty cycle required to achieve the converter's desired output voltage and power. The polynomial model is simple and easy to implement in any form of a digital signal controller (DSC). Normalized parameters are used to widen the operational range and generalize the model. This also offers the essential protection against current and voltage spikes. The work in progress depicts the specific procedures involved in developing a polynomial model. The normalized equations provide a graphical description of the static model, from which the graphical representation of the polynomial are derived. Hence, polynomial equations are obtained. This paper describes the PRC static model, how to convert it to a polynomial model, how to validate it with MATLAB-Simulink, how to program F28335 using Simulink, and how to use it in practice.

**Keywords:** Code Composer Studio; duty; embedded coder; MATLAB; parallel resonant converter (PRC); polynomial model; Simulink; static model; switching frequency

## 1. Introduction

At present, in modern technologies, high voltage is applied in various domains; high voltage levels (tens of KV) are required in a wide range of applications: medical, industrial, environmental, measurement, etc. [1–3]. Due to high efficiency and power density, the use of resonant converters has drastically increased in multiple applications [4–7]. Various power conversion systems have been suggested for decades, as ways of supplying high voltages [8–12]. PRC and PRC-LCC are the most common topologies in high-voltage applications, since, with the inclusion of a high-voltage transformer, they implicate the parasitics as part of the resonant tank [13]. By applying the phase-shifted PWM, it is easier to operate the converter shown in Figure 1 in optimum mode [14], as illustrated in Figure 2. As is well known, in order to operate the converter [14–16], the frequency and duty are modified in such a way that the output voltage and power requirements are achieved. As the duty and frequency are variable, a couple of combinations of power and voltage at the output of the converter are produced and keep the converter operating in the optimum mode. However, at every desired value of output voltage and power, the corresponding frequency and duty have to be pre-calculated [15–19]. The last is a complex process; it renders the implementation more sophisticated. Hence, any algorithm computing it in any

type of DSC is of capital interest. This paper proposes a polynomial model to avoid the static model's complexity. It was first suggested in [15] on a PRC-LCC topology. In [15], a complete model of a parallel resonant converter LCC has been exposed, based on a model that is based on a harmonic approach; the equations and the model used actually are not an approach, and one parameter (frequency) is expressed as a polynomial. In this work, the normalized model is obtained; then, the polynomial equations (frequency and duty cycle) are obtained, and finally, all the steps needed to upload to a digital signal controller are obtained as well. This results in an easy, small, and quick program to implement in the DSC. This work considers the static model of the PRC in Figure 1 as detailed in [14], and limits itself to the optimum mode of operation to elaborate the equivalent polynomial model.

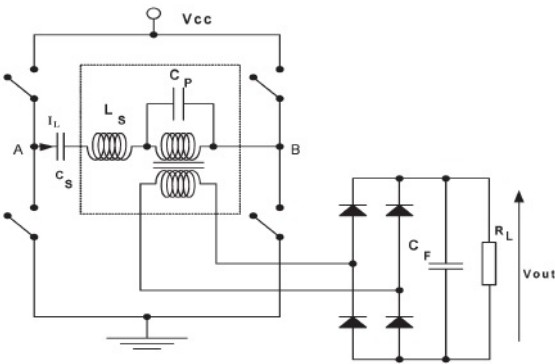

**Figure 1.** Parallel resonant converter [14].

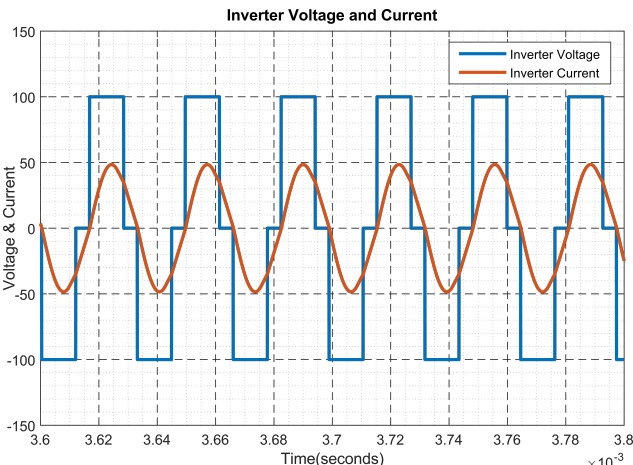

**Figure 2.** Output of inverter in Figure 1 in terms of voltage and current in optimum mode of operation.

From Figure 1, the $L_s$ and $C_p$ reactive elements are included in the transformer parasitic elements; these parasitics are crucial in high-voltage applications [14–17]. For the ease of application and on a wide range, the normalized parameters are used. Hence, Equations (1) and (2) illustrate the respective base and normalized parameters for the indicated PRC, in which $\hat{f}$, $\hat{v}$, $\hat{\Omega}$, and $\hat{p}$ are the normalized frequency, normalized output voltage, normalized load, and normalized power, respectively.

$$V_{Base} = V_{CC} \quad Z_{Base} = \sqrt{\frac{L_S}{C_P}} \quad P_{Base} = \frac{V_{CC}^2}{Z_{Base}}$$
$$f_{Base} = \frac{1}{2\cdot\pi\sqrt{L_s\cdot C_P}} \quad \omega_{Base} = \frac{1}{\sqrt{L_S\cdot C_P}} \tag{1}$$

Based on Equation (1), the normalized parameters are illustrated as follows:

$$\hat{v} = \frac{V_{out}}{V_{CC}} \quad \hat{p} = \frac{P_{out}}{P_{Base}} \quad \hat{\Omega} = \frac{R_L}{Z_{Base}} \quad \hat{f} = \frac{f}{f_{Base}} \tag{2}$$

Figure 2 shows the targeted output of the inverter, where the optimum mode operation is observed.

Figure 3 indicates the complete process for developing a polynomial model and its respective DSC control codes from the static model of the PRC. The last includes the static model, its transformation to a polynomial model, control code generation to DSC, and control wave forms generation for the power converter. The mentioned process is illustrated below:

- First, the static model is developed in MathCad and its graphical representations are subsequently illustrated [14]. To extend the operational span and for ease of analysis, the normalized parameters are used. However, this model is complex in that it renders the implementation sophisticated;
- Second, the polynomial model is suggested [15]. It is an alternative to simplify the model and ease the implementation mostly in DSC. The polynomial model is developed by tuning the static model's graphical representation in such away to generate the polynomic form through the numerical data-fitting method. The last gives the equivalent polynomic expression in graphic representation form;
- Third, the polynomial model is processed in MATLAB-Simulink to produce the program used for simulation and validation. The simulation is run for the validation of the polynomial model and to confirm the accuracy of the control codes' output;
- Fourth, the DSC is configured to Simulink and the control codes are generated. They make the control program that fits in a DCS and give the adequate control waveforms to fire the power converter switches (IGBTs);
- Finally, the experimental results are used to confirm the polynomial model's practicability and implementation feasibilities.

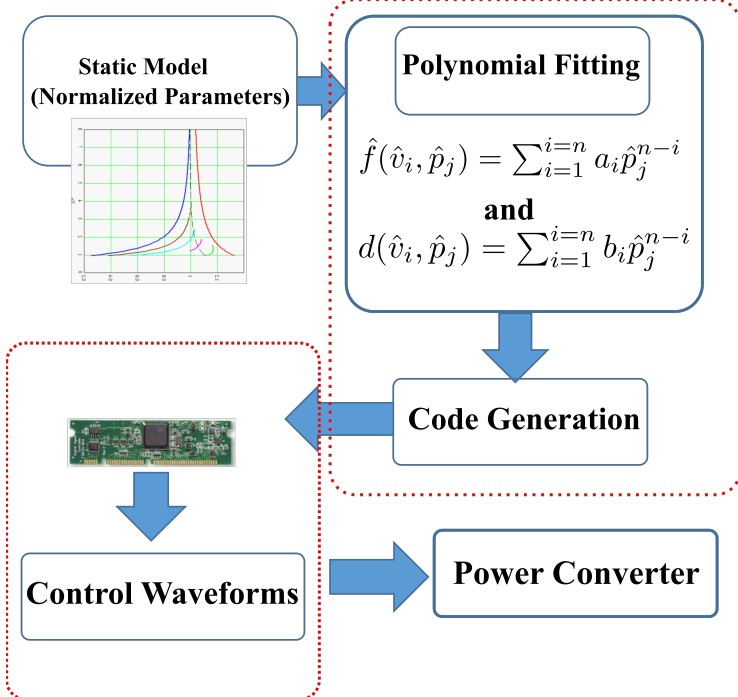

**Figure 3.** Illustration of the various stages of the project.

## 2. Grapical Representation of PRC Static Model

The static model normalized equations are detailed in [14]. In those equations, the normalized output voltage $\hat{v}$ is expressed as a function of the normalized frequency $\hat{f}$ in presence of a variable duty cycle $d$. Meanwhile, the normalized output power $\hat{p}$ is illustrated as a function of the normalized output voltage in presence of a variable duty cycle $d$.

$$\hat{v} = F(\hat{f}, d) \quad and \quad \hat{p} = F(\hat{v}, d) \tag{3}$$

Equation (3) is represented graphically in Figure 4 to illustrate its behaviors. Figure 4a,b shows the normalized output voltage of the model with respect to the change in normalized switching frequency, and in presence of the suitable value of the duty, while Figure 4c illustrates the normalized power as a function of normalized output voltage in presence of the respective duty values. The mentioned graphs highlight the process of establishing the necessary duty and switching frequency for achieving the targeted output voltage and power. It is a long and complex process that groups the implementation mostly onto low-cost DSCs. Recalling that the target is to determine an appropriate duty and switching frequency for every given output voltage and power, Equation (3) is modified to produce an appropriate normalized frequency and duty for each level of output normalized voltage and normalized power, as shown in Equation (4). For simplification of the task, the static model graphical representation is made in such a way to easily generate the duty and switching frequency once the power and output voltage are provided.

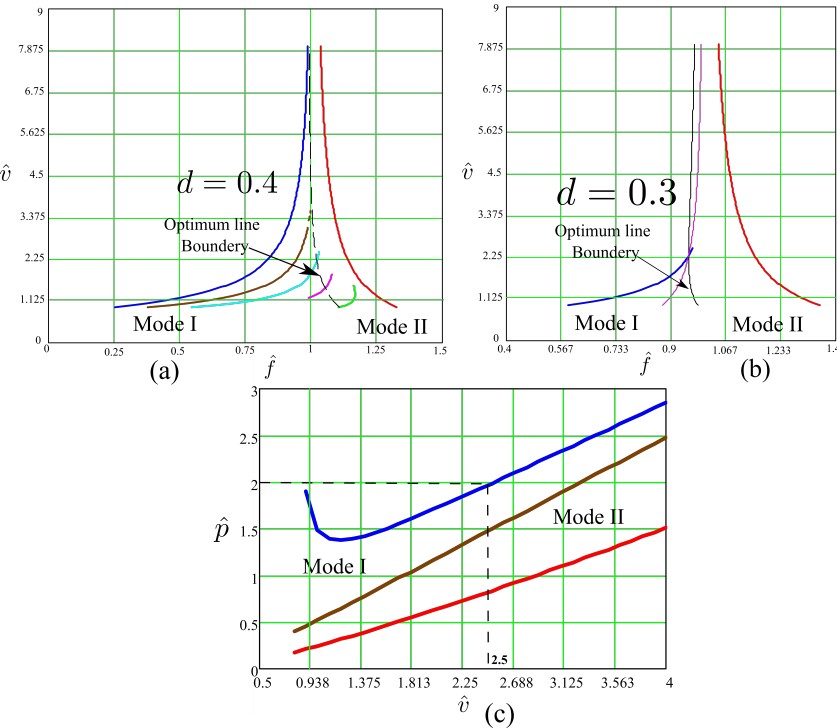

**Figure 4.** Graphical illustration of Equation (3); where (**a**) and (**b**) show the normalized output voltage vs normalized frequency for the duty $d = 0.4$ and $d = 0.3$ respectively. While (**c**) show the normalize power vs normalized output voltage.

The ideal power switches are used on the topology in Figure 1 to extract the graphs illustrated in Figure 4. This assumption does not significantly affect the final results. However, more complex models will be easily integrated in future work [20–22], in such a way to obtain more accuracy and efficiency. The static model's graphical representation in Figure 4a–c are rectified in a way to give the values of the duty and switching frequency at any set value of output voltage and power. Hence, the new form of graphical representation

is illustrated in Figure 5. In the last, the switching frequency and duty are functions of output voltage and power.

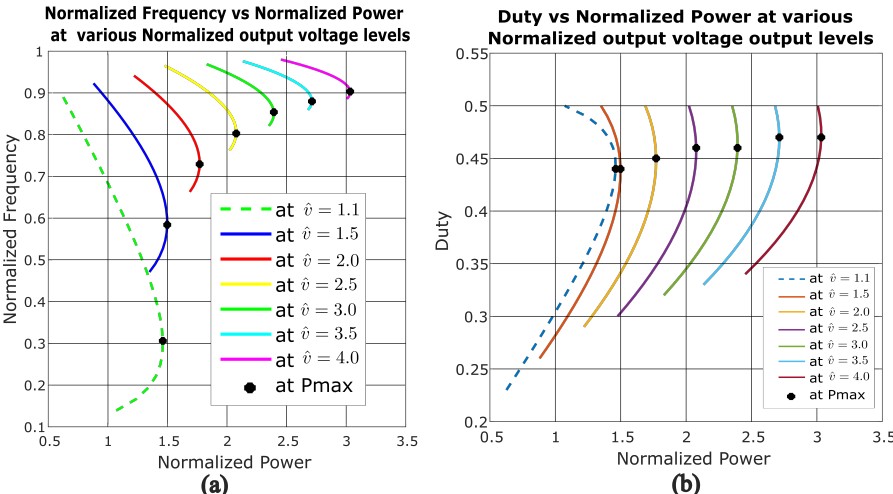

**Figure 5.** Graphical illustration where: (**a**) shows normalized Frequency vs normalized power, (**b**) shows duty vs normalized power. In presence of various level of normalized output voltage.

Figure 5a indicates the normalized frequency as a function of the normalized output power, in presence of the normalized output voltage. Hence, the normalized frequency is a function of both the normalized output voltage and normalized output power. In Figure 5b, the duty as a function of normalized power, in presence of normalized voltage, is illustrated; it implies that the duty is a function of both the normalized output voltage and the normalized power. Figure 5 can be expressed using Equation (4).

$$\hat{f} = F(\hat{v}, \hat{p}) \quad and \quad d = F(\hat{v}, \hat{p}) \tag{4}$$

Taking Figure 5 into account, the power reaches its maximum at each level of voltage, with a specific frequency or duty, and then begins to fall. Figure 6 depicts the power as a function of the frequency and duty cycle. The power at point **A** has the same value as the power at point **B**. Point **B**, on the other hand, comes after the maximum power point **Pmax**. Similarly, the value of the power at point **C** is the same as the value of the power at point **D**. After the maximum power point **Pmax**, point **D** is attained. The power beyond the maximum level **Pmax** is ignored by the design in place for power maximization at each frequency, duty, and voltage level. Assuming that Equation (4) is made of two polynomials for switching frequency and duty, respectively, the next step is to find their respective coefficients. This process will result in a set of polynomials of the *n*th order. The last have to be simple and make the control philosophy to run the control topology in Figure 1 on the DSC and in the optimum mode of operation.

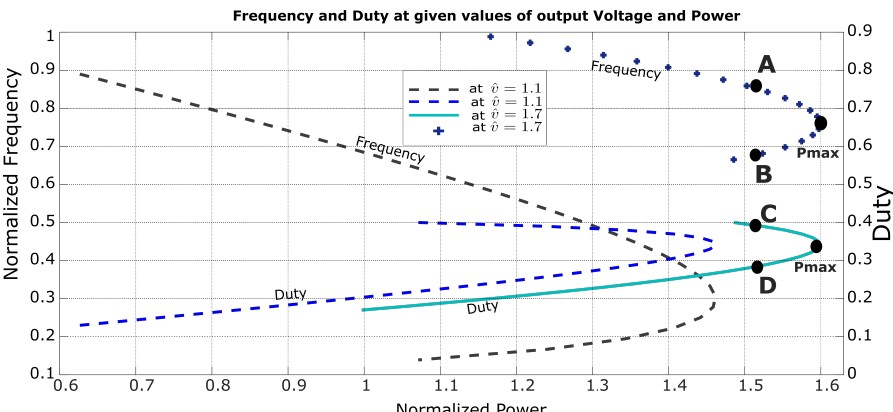

**Figure 6.** Illustration of normalized frequency vs. normalized power and duty vs. normalized power in presence of normalized voltage with the related behaviors. A and B show the point of same power on a value of frequency, while C and D show the point of same power on a value duty.

## 3. Polynomial Model of Parallel Resonant Converter

Having concurred with the assumption that Equation (4) is made of two polynomials, the next step is to find the adequate coefficients for each polynomial. In this case, numerical data fitting is applied on Figure 7a,d to retrieve the suitable coefficients for the polynomials. Recalling that, at each level of voltage, the power beyond the maximum is ignored, Figure 7 is a portion of Figure 5 at the normalized output voltage $\hat{v} = 1.1$ and $\hat{v} = 1.7$, respectively.

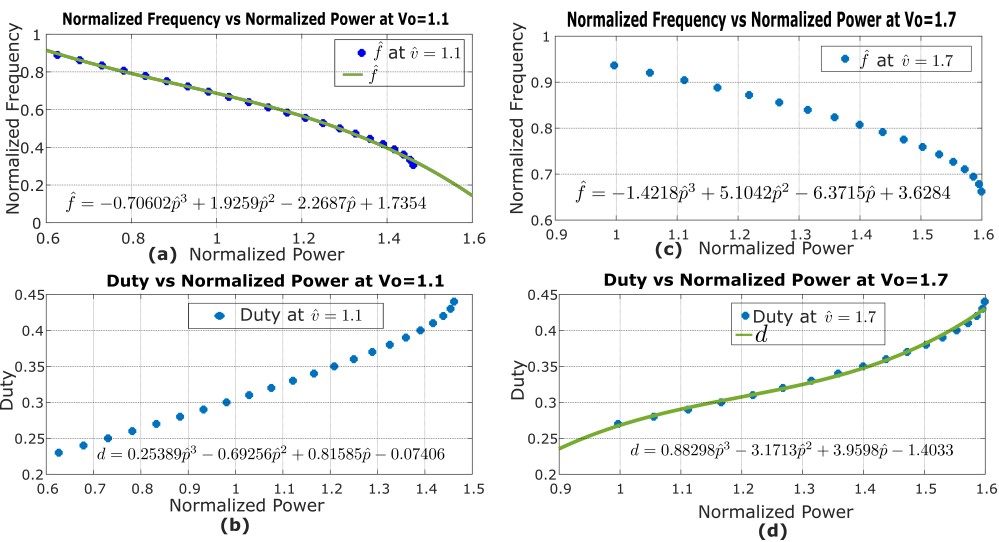

**Figure 7.** Data-fitting application on frequency and duty in presence of $\hat{v} = 1.1$ and $\hat{v} = 1.7$. (**a**,**b**) show data fitting for frequency and duty at $\hat{v} = 1.1$, while (**c**,**d**) show data fitting for frequency and duty at $\hat{v} = 1.7$.

The data-fitting method is applied to each specific voltage level plot for both frequency and duty, as illustrated in Figure 7, to obtain the equivalent polynomic expression. Various fittings were tried, including each fitting above the cubic over-fitted and each fitting below the cubic under-fitted. Therefore the accurate fitting was realized at the cubic level. This process resulted in the coefficients shown in Tables 1 and 2 for frequency and duty, respectively.

Tables 1 and 2 illustrate the fittings from different frequencies and duties, at various levels of voltage and power. The coefficient $a_i$ is for frequency, while $b_i$ is for duty. Hence, Equation (4) results in Equations (5) and (6) in polynomic form:

$$\hat{f}(\hat{v}, \hat{p}) = a_0 \hat{p}^3 + a_1 \hat{p}^2 + a_2 \hat{p} + a_3 \tag{5}$$

for frequency, and:

$$d(\hat{v}, \hat{p}) = b_0\hat{p}^3 + b_1\hat{p}^2 + b_2\hat{p} + b_3 \tag{6}$$

for duty.

The coefficients $a_i$ and $b_i$ in Tables 1 and 2, respectively, are functions of the output voltage $\hat{v}$ and polynomials as well. They are in the form illustrated in Equation (7). Therefore, their respective coefficients are computed in turn. The data in Tables 1 and 2 are graphically represented and data fitting is applied, as shown in Figure 8a.

$$a_i = F(\hat{v}) \quad and \quad b_i = F(\hat{v}) \tag{7}$$

However, data fitting results in big outliers once applied to coefficients $a_i$ and $b_i$, as in Figure 8a, which were immediately ignored. Therefore, an alternative was to apply B-spline [23–26]. The last is part of numerical analysis, which is mostly applied in curve-fitting and numerical differentiations of data. By applying B-spline to data in Tables 1 and 2, the coefficients for polynomials $a_i$ and $b_i$ are retrieved. The obtained coefficients are piece-wise linear at each output voltage level, as shown in Figure 8b.

**Table 1.** The table illustrates various coefficients of different fittings on frequency plots, at specific levels of voltage. $a_i$: coefficients for frequency fitting.

| $\hat{v}$ | $a_0$ | $a_1$ | $a_2$ | $a_3$ |
|---|---|---|---|---|
| 1.0 | 0 | 0 | −0.61102 | 1.222 |
| 1.1 | −0.70602 | 1.9259 | −2.2687 | 1.7354 |
| 1.2 | −1.288 | 3.6637 | −3.9182 | 2.2918 |
| 1.3 | −1.3247 | 3.9212 | −4.2784 | 2.4819 |
| 1.4 | −1.5175 | 4.6649 | −5.1259 | 2.8205 |
| . | . | . | . | . |
| . | . | . | . | . |
| . | . | . | . | . |
| 3.6 | −0.59499 | 4.2475 | −10.184 | 9.1596 |
| 3.7 | −0.74666 | 5.5661 | −13.909 | 12.606 |
| 3.8 | −0.65556 | 4.9904 | −12.737 | 11.858 |
| 3.9 | −0.57735 | 4.486 | −11.689 | 11.175 |
| 4.0 | −0.51036 | 4.0457 | −10.758 | 10.559 |

**Table 2.** The table illustrates various coefficients of different fittings on duty plots, at specific levels of voltage. $b_i$: coefficients for duty fitting.

| $\hat{v}$ | $b_0$ | $b_1$ | $b_2$ | $b_3$ |
|---|---|---|---|---|
| 1.0 | 0 | 0 | 0.19449 | 0.11102 |
| 1.1 | 0.25389 | −0.69256 | 0.81585 | −0.07406 |
| 1.2 | 0.51726 | −1.4713 | 1.5735 | −0.32038 |
| 1.3 | 0.58838 | −1.7417 | 1.9003 | −0.45238 |
| 1.4 | 0.74316 | −2.2865 | 2.5136 | −0.68055 |
| . | . | . | . | . |
| . | . | . | . | . |
| . | . | . | . | . |
| 3.6 | 0.89738 | −6.408 | 15.368 | −12.018 |
| 3.7 | 1.1575 | −8.6289 | 21.562 | −17.693 |
| 3.8 | 1.048 | −7.9784 | 20.364 | −17.059 |
| 3.9 | 0.95176 | −7.3958 | 19.273 | −16.475 |
| 4.0 | 0.86674 | −6.872 | 18.276 | −15.935 |

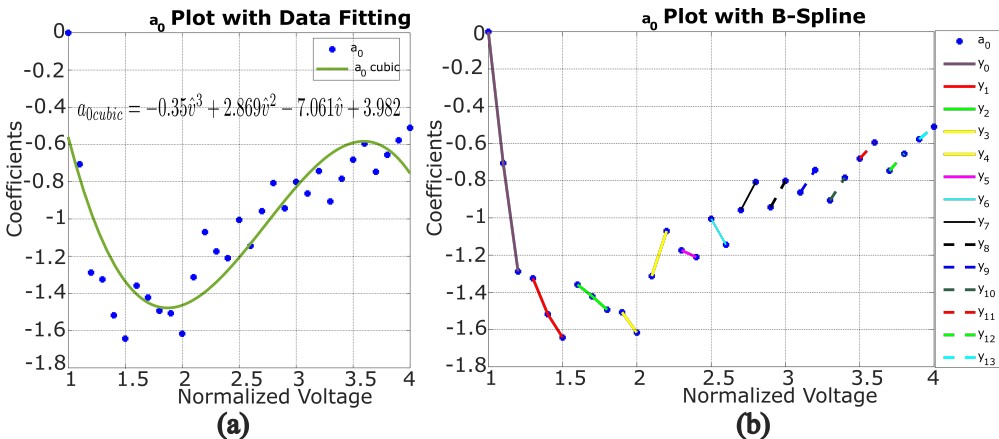

**Figure 8.** Illustration of $a_0$ plotting, data fitting and B-spline application; where (**a**) shows data fitting while (**b**) shows its equivalent B-spline representation.

The application of B-spline results in the linear expressions displayed in Equation (8). Tables 3 and 4 give the detailed coefficients obtained through B-spline application from data in Tables 1 and 2, respectively, for Equation (8).

**Table 3.** This table shows the specific coeffiecients retrieved from the coefficients in Table 1 after B-spline.

| $\hat{v}$ | $a_0$ | | $a_1$ | | $a_2$ | | $a_3$ | |
|---|---|---|---|---|---|---|---|---|
| | $\alpha_0$ | $\lambda_0$ | $\alpha_1$ | $\lambda_1$ | $\alpha_2$ | $\lambda_2$ | $\alpha_3$ | $\lambda_3$ |
| 1.0–1.2 | −6.4400 | 6.4193 | 18.3190 | −18.2870 | −16.5360 | 15.9240 | 5.3490 | −4.1342 |
| 1.3–1.5 | −1.5900 | 0.7310 | 7.2865 | −5.5462 | −9.5360 | 8.1538 | 4.2220 | −3.0346 |
| 1.6–1.8 | −0.6750 | −0.2770 | 5.6585 | −4.4915 | −10.4860 | 11.3930 | 5.8700 | −6.3060 |
| 1.9–2.0 | −1.0950 | 0.5737 | 8.3280 | −9.8350 | −16.5230 | 23.2470 | 10.0220 | −14.3620 |
| 2.1–2.2 | 2.4240 | −6.4033 | −9.1850 | 24.9940 | 11.4450 | −32.4880 | −4.4950 | 14.6110 |
| 2.3–2.4 | −0.3650 | −0.3347 | 3.7960 | −3.1700 | −9.1530 | 12.1060 | 6.5700 | −9.3089 |
| 2.5–2.6 | −1.3930 | 2.4775 | 10.2110 | −20.4600 | −22.7860 | 48.3120 | 16.1910 | −34.5450 |
| 2.7–2.8 | 1.5088 | −5.0318 | −7.1010 | 24.4050 | 10.8790 | −39.0250 | −5.2610 | 21.150 |
| 2.9–3.0 | 1.4187 | −5.0571 | −7.1910 | 26.4110 | 11.8340 | −45.3510 | −6.1590 | 26.1780 |
| 3.1–3.2 | 1.2073 | −4.6060 | −6.4090 | 25.2430 | 11.0000 | −45.3490 | −5.9280 | 27.2370 |
| 3.3–3.4 | 1.2193 | −4.9297 | −6.9400 | 28.9530 | 12.7600 | −55.6680 | −7.3800 | 35.5000 |
| 3.5–3.6 | 0.8654 | −3.7104 | −5.1330 | 22.7260 | 9.8200 | −45.5360 | −5.8950 | 30.3820 |
| 3.7–3.8 | 0.9110 | −4.1174 | −5.7570 | 26.8670 | 11.7200 | −57.2730 | −7.4800 | 40.2820 |
| 3.9–4.0 | 0.6699 | −3.1900 | −4.4030 | 21.6580 | 9.3100 | −47.9980 | −6.1600 | 35.1990 |

**Table 4.** This table shows the specific coeffiecients retrieved from the coefficients in Table 2 after B-spline.

| $\hat{v}$ | $b_0$ | | $b_1$ | | $b_2$ | | $b_3$ | |
|---|---|---|---|---|---|---|---|---|
| | $\beta_0$ | $\phi_0$ | $\beta_1$ | $\phi_1$ | $\beta_2$ | $\phi_2$ | $\beta_3$ | $\phi_3$ |
| 1.0–1.2 | 2.5863 | −2.5879 | −7.3565 | 7.3709 | 6.8951 | −6.7233 | −2.1570 | 2.2782 |
| 1.3–1.5 | 1.4197 | −1.2530 | −5.5730 | 5.5074 | 6.9235 | −7.1266 | −2.8196 | 3.2310 |
| 1.6–1.8 | 1.0212 | −0.8488 | −5.6690 | 6.4327 | 9.2140 | −11.6320 | −4.7130 | 6.5631 |
| 1.9–2.0 | 1.4800 | −1.7409 | −8.8670 | 12.5890 | 15.9650 | −24.5400 | −9.0870 | 14.8890 |
| 2.1–2.2 | −1.4155 | 4.0245 | 4.9870 | −15.0450 | −5.5260 | 18.3800 | 1.7760 | −6.8231 |
| 2.3–2.4 | 0.9160 | −1.0600 | −6.2970 | 9.5256 | 13.0150 | −21.9590 | −8.5190 | 15.5710 |
| 2.5–2.6 | 1.8546 | −3.6424 | −12.6150 | 26.5220 | 27.0250 | −58.9940 | −18.6480 | 42.0000 |
| 2.7–2.8 | −1.2309 | 4.3582 | 5.4290 | −20.3120 | −7.4930 | 30.6620 | 3.0070 | −14.2730 |
| 2.9–3.0 | −1.2665 | 4.7769 | 6.0270 | −23.9850 | −8.970 | 38.9310 | 3.9060 | −19.6160 |
| 3.1–3.2 | −1.1662 | 4.7107 | 5.7960 | −24.7890 | −8.9300 | 41.9620 | 3.9460 | −21.9280 |
| 3.3–3.4 | −1.2710 | 5.4262 | 6.7870 | −30.6250 | −11.2500 | 55.5640 | 5.3800 | −31.2600 |
| 3.5–3.6 | −0.9676 | 4.3807 | 5.3780 | −25.7690 | −9.2600 | 48.7040 | 4.5600 | −28.4340 |
| 3.7–3.8 | −1.0950 | 5.2090 | 6.5050 | −32.6970 | −11.9800 | 65.8880 | 6.3400 | −41.1510 |
| 3.9–4.0 | −0.8502 | 4.2675 | 5.2380 | −27.8240 | −9.9700 | 58.1560 | 5.4000 | −37.5350 |

Equation (7) results in two capital polynomial coefficients, for frequency and duty, respectively, as follows:

$$a_i(\hat{v}) = \alpha_i \hat{v} + \lambda_i \quad and \quad b_i(\hat{v}) = \beta_i \hat{v} + \phi_i \tag{8}$$

Equations (5) and (6) provide the frequency and duty that are able to run the topology on Figure 1 in its optimum mode at a specified level of voltage, power, and load. Combining Equations (5), (6) and (8), the general Equation (9) for frequency and duty, respectively, is obtained. They can fit in any DSC and generate the suitable control signals.

$$\hat{f}(\hat{v}_i, \hat{p}_j) = \sum_{i=0}^{i=n} a_i \hat{p}_j^{n-i} \quad and \quad d(\hat{v}_i, \hat{p}_j) = \sum_{i=0}^{i=n} b_i \hat{p}_j^{n-i} \tag{9}$$

## 4. Model Building and Simulations

In order to validate the polynomials shown in Equation (9). The model in Figure 1 was built in MATLAB-Simulink R2021a. The polynomials were applied to the mentioned model to confirm the optimum mode of operation. The model validation considers output voltage and power as independent variables, while the switching frequency and duty are dependent. The last is illustrated in Equation (9). Each level of output voltage and power corresponds not only to frequency and duty but also to the load. Hence, the load is dynamic in this case, and governed by Equation (10), which is a polynomial in turn. The control model was built in MATLAB-Simulink as well. It is made of a MATLAB function block to hold Equation (9). This block accepts two inputs (normalized output voltage and normalized power) and generates two outputs (normalized frequency and duty). The two last outputs are used to obtain the required period and phase shift that run the PWM generator. The last, in turn, generates the switching signals that run the IGBTs on the model in Figure 1. The control model is illustrated in both Figures 9 and 10. Both the power and control model are run in MATLAB-Simulink for validation of the polynomials before going for practical implementation.

$$\hat{\Omega} = F(\hat{v}_i, \hat{p}_j) \tag{10}$$

Practically, the load is not always variable; it is fixed in most cases. Therefore, the simulation will be conducted in two ways: two-inputs model simulation and one-input model simulation.

### 4.1. Two-Inputs Model Simulations

The normalized voltage and normalized power are the capital. They change the independent variables to the polynomials in Equation (9). Hence, they are the two inputs for the model in Figure 9. They are fed to the polynomials' block. The last holds Equation (9); they are written in the form of control codes with the necessary protection and generate from them the necessary duty and frequency. The frequency is converted into an equivalent period and it is fed to PWM generator for PWM signal generation. The duty is used to calculate the shifting time; this is fed to the phase shift block and helps generate the phase-shifted PWM signals.

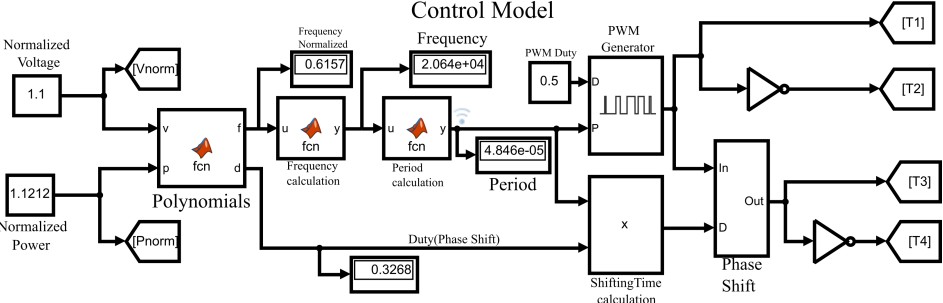

**Figure 9.** Illustration of Simulink model with normalized voltage and normalized power as inputs.

$$\hat{\Omega} = \frac{\hat{v}^2}{\hat{p}} \tag{11}$$

The normalized load at each level of voltage and power is given by Equation (11). The simulation component values used in the simulation are from the prototype components' values. They are indicated in Table 5.

Equations (5) and (6) were simulated in Simulink. The parameters in Table 5 were used. The input voltage is $V_{in} = 100$ V, while the normalized value of the output voltage is $\hat{v} = 2.5$. Then, the expected value of the output voltage is $V_{out} = 250$ V. If the normalized power $\hat{p} = 1.83177461$ is considered, then the required frequency and duty are obtained by applying Equations (5) and (6) to the model in Figure 3. The frequency $\hat{f} = 0.9086$ and the duty $d = 0.3566$ were obtained and applied in the Simulink model. The load value is given by Equation (11) and the value by $\hat{\Omega} = 3.412$. The results of the simulation are shown in Figure 10. In this case, the optimum mode is fulfilled along the domain of the functions in Equation (9).

**Table 5.** The parameters used to simulate the model in Figure 9 with dynamic load.

| Item | Value | Unit |
|------|-------|------|
| $V_{in}$ | 100 | V |
| $L_s$ | 34.232 | uH |
| $C_p$ | 658.7 | nF |
| $C_f$ | 30 | uF |
| $Z_{Base}$ | 7.209 | Ω |
| $f_{Base}$ | 33.527 | kHz |

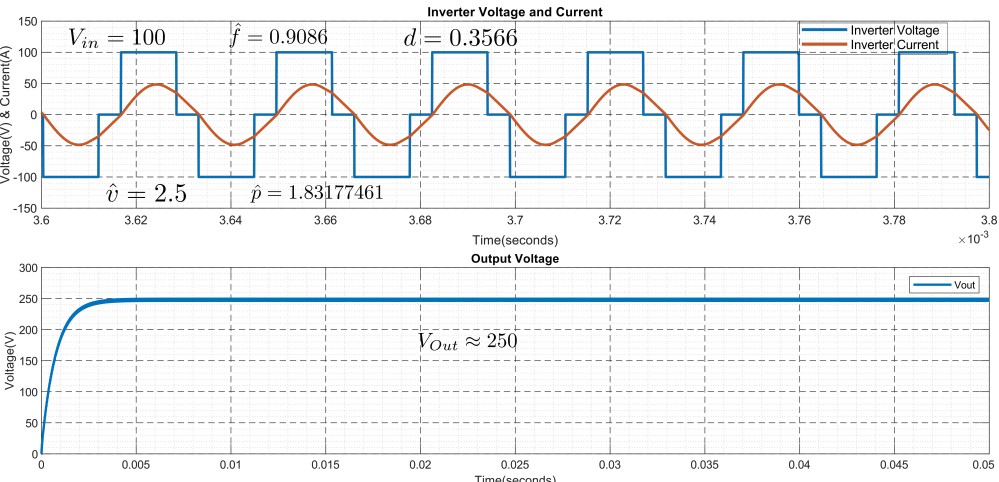

**Figure 10.** The voltage and current output of the converter with normalized output voltage and normalized power as inputs.

### 4.2. One-Input Model Simulations

In this case, the load is constant. Any change in the output power results in a different level of output voltage. The power is the only input to the control model, the corresponding voltage is illustrated with respect to the fixed load. Except having one input, the rest of the model components fit the model description in Section 4.1 Figure 11.

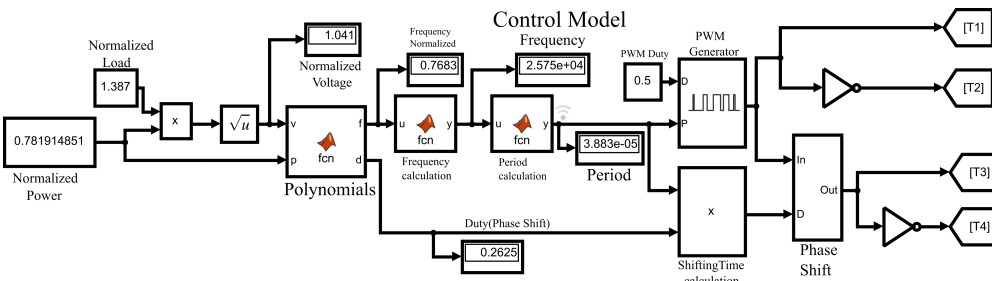

**Figure 11.** One-input Simulink control model.

Therefore, having the load as a fixed parameter and the power as an input, the corresponding voltage is given by Equation (12).

$$\hat{v} = \sqrt{\hat{p} * \hat{\Omega}} \tag{12}$$

The simulation parameters are given in Table 5.

The power and the load are $\hat{p} = 0.7819$ and $\hat{\Omega} = 1.387$, respectively; hence, the voltage is $\hat{v} = 1.0414$. In this case, the frequency and duty are $\hat{f} = 0.76827$ and $d = 0.2625$, respectively. Figure 12 displays the simulation results. In case of a fixed load, the optimum mode is fulfilled at a fixed range of voltage. Hence, $\hat{\Omega} = 1.387$ runs from $\hat{v} = 1.00664$ to $\hat{v} = 1.441706$.

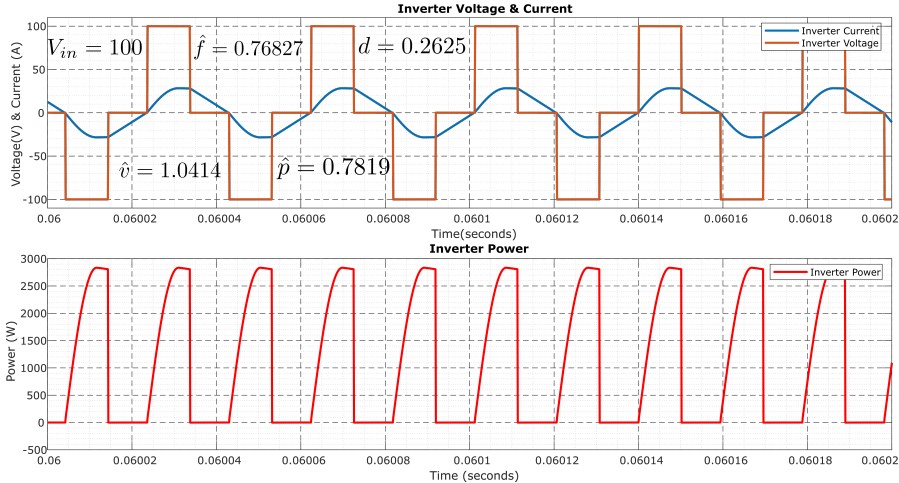

**Figure 12.** Inverter voltage, current, and power at fixed load

## 5. DSC Configuration

Due to the aforementioned operations, the indicated controls cannot be easily implemented. The DCS is key, as it is has the computational capability; this work will take advantage of a Texas Instrument (TI) DSC. The last is a TMS320F28335 Peripheral Explorer Kit with high-performance static CMOS technology, a high-performance 32-bit CPU, fast interrupt response processing, on-chip memory, and many more advantages [27]. The shifted pulse-width modulation (PWM) switching signals were generated on a TMS320F28335 Peripheral Explorer Kit using MATLAB-Simulink. In fact, the TMS320F28335 is a Texas Instrument micro-controller, programmed through a Code Composer Studio development environment (CCS). CCS supports C or C++ language. Even if they are common to most control engineers, the mentioned languages are time-consuming and prone to several errors; hence, it is tiresome to code the signals accurately. However, those challenges are responded to by using model-based programming through MATLAB-Simulink. The complete control algorithm is completed in MATLAB-Simulink and, later on, the PWM codes for the F28335 are generated. Some necessary steps are crucial to configure a Peripheral Kit to MATLAB-Simulink. MATLAB-Simulink, with some necessary add-ons such as "Embedded Coder",

"Simulink Coder", and "MATLAB Coder", together with CCS, have to be installed on the PC. In this case, MATLAB R2021a, with the mentioned add-ons, together with CCS 7.3.0, are installed on the PC. Hence, the target hardware (TI Delfino F2833x) is configured. The model is built in Simulink, through which the PWM codes are generated and downloaded to a TMS320F28335 Peripheral Explorer Kit. The codes for backup or any other analysis are retrieved in project form through CCS. Equation (9) is programmed in the F28335 using a MATLAB function block; they are quick and fast-response polynomials. In the same coding is included the protection of the system, in case of mistakes when entering values.

Figure 13 mentions the model-based program to generate PWM for control purposes. The Hex-Encoder on the F28335 serves as the input, as it accepts 16 inputs at different input steps. The Hex-Encoder is linked to four general-purpose input–outputs (GPIO). The last are GPIO15, GPIO14, GPIO13, and GPIO12, in order of most to least significant bit, respectively. They input 16 different values of power with the help of a lookup table. Taking into consideration the fixed load, Equation (12) gives the voltage. Both voltage and power are fed to polynomial block. The last hosts the Equation (9)-related codes. They generate, in turn, the corresponding frequency and duty. The frequency is converted to the corresponding period according to Equation (13). The period is fed to both ePWM1 and ePWM2 at input "T" for PWM signal generation for both converter legs. A half of the period is fed to both ePWM1 and ePWM2 on inputs "WA" and "WB" to maintain a 50% duty cycle for each switching signal. The product of period and duty gives the adequate time-shift between the converter legs, and it is fed to ePWM2 on input "PHS". The PWM-switching signals at adequate frequencies with the necessary duty shifting are generated. The ePMW1A and ePWM1B output the switching signal for the first leg, while ePWM2A and ePWM2B give the shifted switching signal for the second leg.

$$T = \frac{f_{clk}}{2 * f_{sw}} \tag{13}$$

From Equation (13), $T$ is the period, $f_{clk}$ is the F28335 maximum clock frequency, and $f_{sw}$ is the switching frequency.

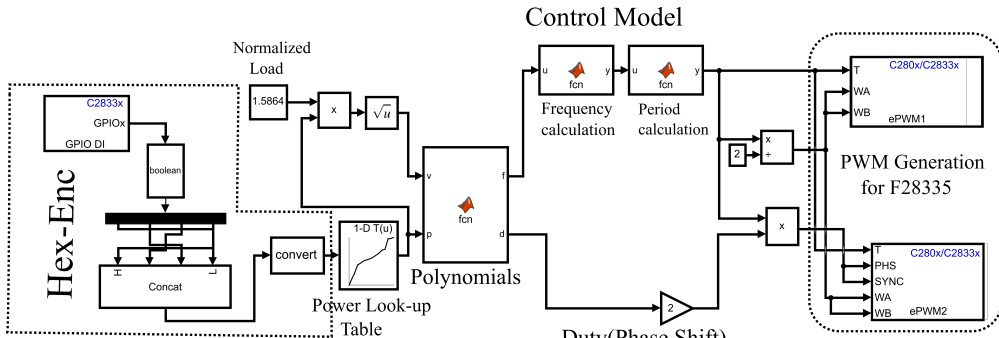

**Figure 13.** PWM generation model in Simulink.

## 6. Experimental Prototype

The topology shown in Figure 1 was used to build the prototype for practical experimentations, with the target of obtaining the model illustrated in Figure 14 and to practically prove the polynomial model. It is a topology made of a personal computer for programming purposes. The F28335 micro-controller is used for hosting the control program and to generate the corresponding switching signals, $S^*_1$, $S^*_2$, $S^*_3$, and $S^*_4$. The drivers are connected to the output of the controller and strengthen the switching signals to a suitable level of $S_1$, $S_2$, $S_3$, or $S_4$ for firing the IGBTs. The embedded IGBTs are used and provide an adequate commutation for the proper output of the prototype. The IGBTs form a converter and link the input voltage to the resonant tank. The last is made of leakage inductance and parallel capacitance connected in parallel. Both these components can be part of the power transformer

as parasitics. The diode rectifier and the filter capacitor output the DC voltage with proper smoothness. Figure 14 presents the steps and components of the prototype.

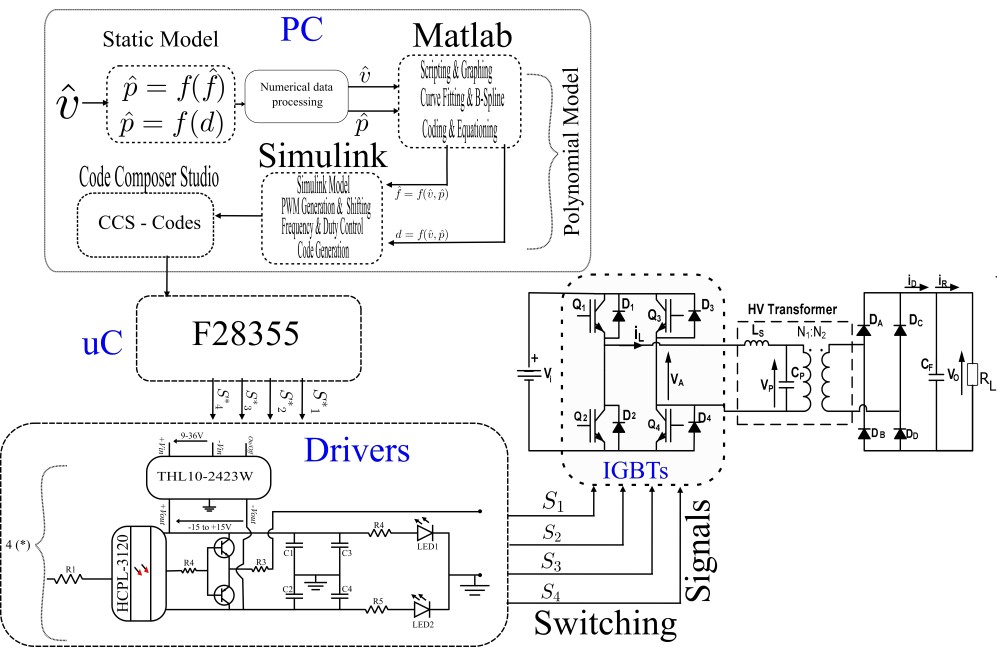

**Figure 14.** Illustration of the complete model of the practical implementation.

### 6.1. Driver

The drivers are signal-strengthening circuits. They are fed from the F28335 microcontroller. They give the level necessary to switch the IGBTs to the switching signals. They replicate into four similar drivers: one driver for each signal, as presented in Figure 15. Each driver is made of an HCPL-3120 opto-coupler to electrically decouple the references of input and output signals. The THL10-2423W is a DC–DC power converter that receives 9 V to 36 V and outputs $-15$ V to $+15$ V. The complete driver receives a switching signal of 0 to 3.3 V and outputs $-15$ V to $+15$ V. Figure 15 shows the driver board. The F4-150R12KS4 is the embedded circuit made of two legs of IGBTs. It is small, compact, and mounted on the driver board, which makes the complete topology small in size.

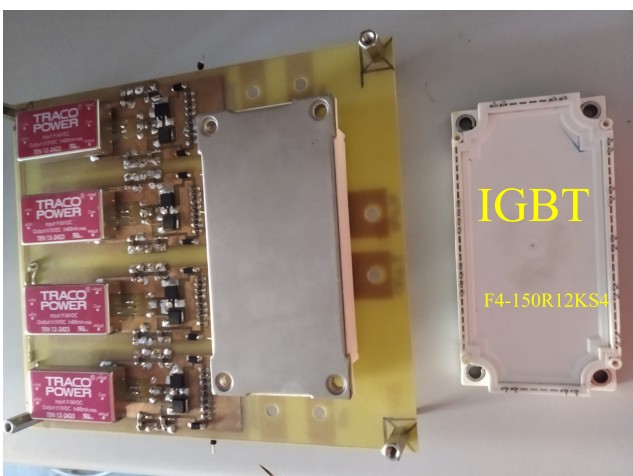

**Figure 15.** Illustration of the driver model.

The prototype can have different behaviors based on the components of the resonant tank. Two scenarios are tested in this work. The first scenario is the low-voltage prototype, where the resonant tank is made of an independent capacitor and inductor without a power

transformer. This is the low-voltage case. The second option considers the resonant tank with an independent inductor together with the power transformer and its parasitics. This is a high-voltage case.

### 6.2. Low-Voltage Test

The prototype was made up of an F28335 TI micro-controller for shifted PWM signals, a driver to strengthen the PWM signals to a required switching level, an F4-150R12KS4-embedded IGBT converter, a resonant tank of $L_s = 34.232$ μH and $C_p = 658.7$ nF, an uncontrolled rectifier, a filter capacitor of 30 μF, and a 10 Ω load.

The prototype was tested on a fixed load of 10 Ω by applying Equation (12) on the program model in Figure 13. The power values in Table 6 were applied.

Table 6 shows the data retrieved on the prototype in Figure 16. The same prototype was run $\hat{v}$ = 1.00664–1.441706 and $\hat{p}$ = 0.730587–1.498569. The resonant frequency of the prototype was $f_r = 33.517$ kHz. The behavior of the prototype at the fixed load is illustrated in Figure 17. As the output voltage increased, the power transfer increased as well. Hence, the switching frequency goes further below resonance, while the duty increases drastically.

Figure 18 shows the behaviors of the prototype in Figure 16 in optimum mode. It was obtained through running the topology in Figure 16 in optimum mode and on the following data: power: $\hat{p} = 0.781915$; normalized load: $\hat{\Omega} = 1.387$. Hence, the voltage becomes $\hat{v} = 1.0414$. At the input voltage $V_{in} = 39.6$ V, the output voltage becomes $V_{out} = 41.24$. Therefore, the obtained frequency and duty are $\hat{f} = 0.76826685 \approx f = 25.75$ kHz and $d = 0.2625$, respectively.

**Table 6.** This table shows different operating points in the design operating window of the prototype in Figure 16 with a load of 10 Ω $\approx \hat{\Omega}$ = 1.387.

| $\hat{v}$ | $\hat{p}$ | $\hat{f}$ | $d$ |
|---|---|---|---|
| 1.00664 | 0.730587 | 0.780201 | 0.2535 |
| 1.041401 | 0.781915 | 0.768267 | 0.2625 |
| 1.074677 | 0.832682 | 0.757526 | 0.2716 |
| 1.120123 | 0.904597 | 0.74589 | 0.2842 |
| 1.150194 | 0.953819 | 0.74052 | 0.2925 |
| 1.172152 | 0.990585 | 0.737835 | 0.2985 |
| 1.30065 | 1.219677 | 0.695766 | 0.3417 |
| 1.316569 | 1.249714 | 0.685622 | 0.3488 |
| 1.332491 | 1.280124 | 0.673986 | 0.3567 |
| 1.348824 | 1.311699 | 0.661157 | 0.3656 |
| 1.359014 | 1.331593 | 0.652206 | 0.3716 |
| 1.372422 | 1.357997 | 0.639675 | 0.3801 |
| 1.389738 | 1.392481 | 0.621476 | 0.3924 |
| 1.403329 | 1.419851 | 0.605663 | 0.4032 |
| 1.412658 | 1.438791 | 0.593729 | 0.4112 |
| 1.441706 | 1.498569 | 0.550467 | 0.4404 |

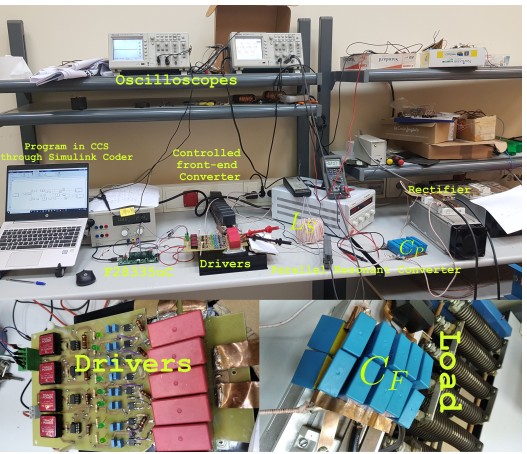

**Figure 16.** Illustration of the low-voltage prototype.

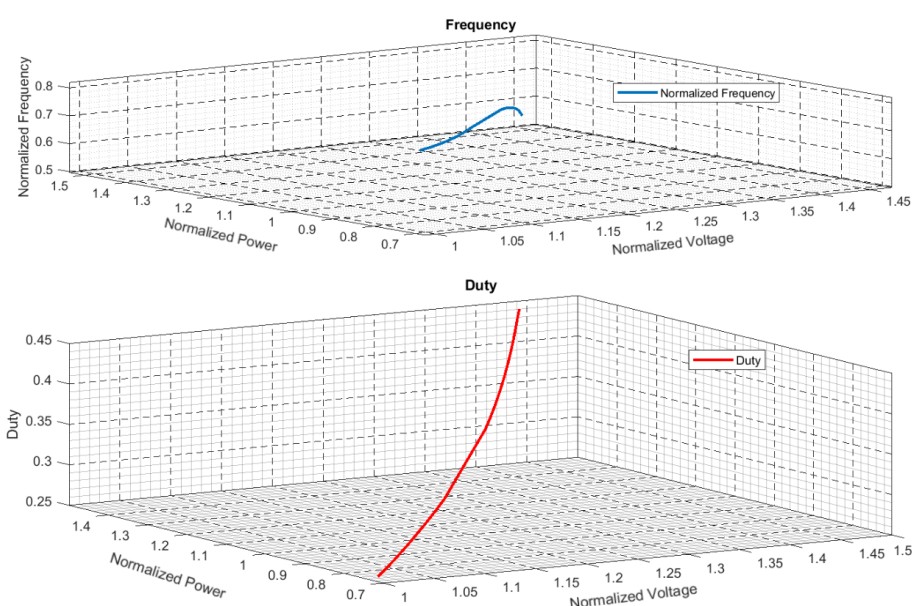

**Figure 17.** Illustration of normalized frequency and duty at a fixed load of $\hat{\Omega} = 1.387$, corresponding to 10 Ω.

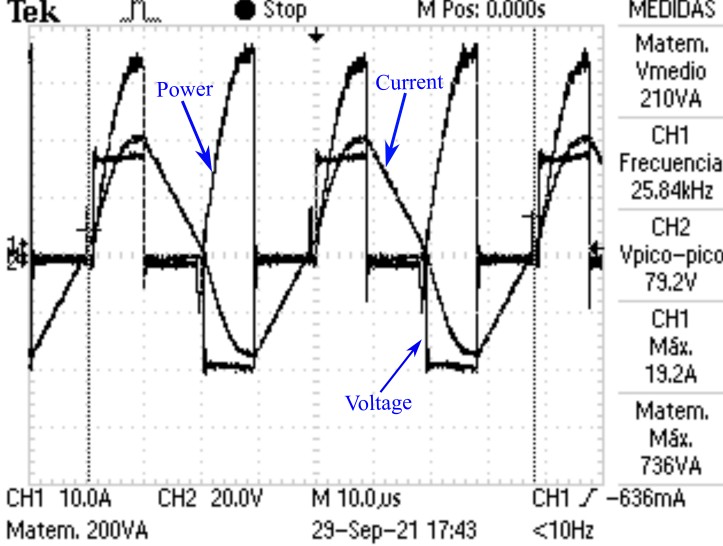

**Figure 18.** Illustration of the prototype output.

### 6.3. High-Voltage Test

The high-voltage prototype was built as illustrated in Figure 19. It uses the same components as the low-voltage prototype, with the exception of the high-voltage transformer; this was added for a higher transformation ratio. The inductor $L_S = 110.34$ µH (with 22.4 µH of transformer leakage inductance) and the parasitic capacitor $C_P = 43$ nF, the load is 57.575 Ω, the resonant frequency of $f_r = 73.067$ kHz, $Z_{base} = 50.6561$ Ω, and $n = 400/14$ is the transformer turn ratio. Transformer oil was used to insulate the transformer and the load for safe runs under high-voltage outputs. The maximum output voltage reached was 4 kV. Figure 19 shows the high-voltage transformer and the load immersed in transformer oil for high-voltage insulation.

The prototype in Figure 19 was run at a fixed normalized load of $\hat{\Omega} = 1.1366$. The model-based program illustrated in Figure 13 runs on various power levels, in a range between $\hat{p} = 0.882826$ and $\hat{p} = 1.498569$. Figure 20 shows the output of the high-voltage

test, where the input voltage is $V_{in} = 94$, the input current $I_{in} = 1$ A, the normalized power $\hat{p} = 0.882826$, the normalized frequency $\hat{f} = 0.6862$, the duty $d = 0.283$, and the output voltage $V_{out} = 2$ kV. The optimum mode is realized.

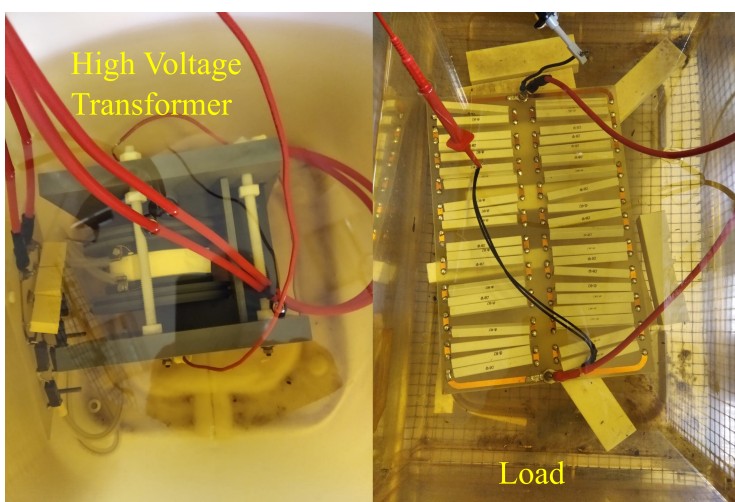

**Figure 19.** Illustration of the high-voltage prototype.

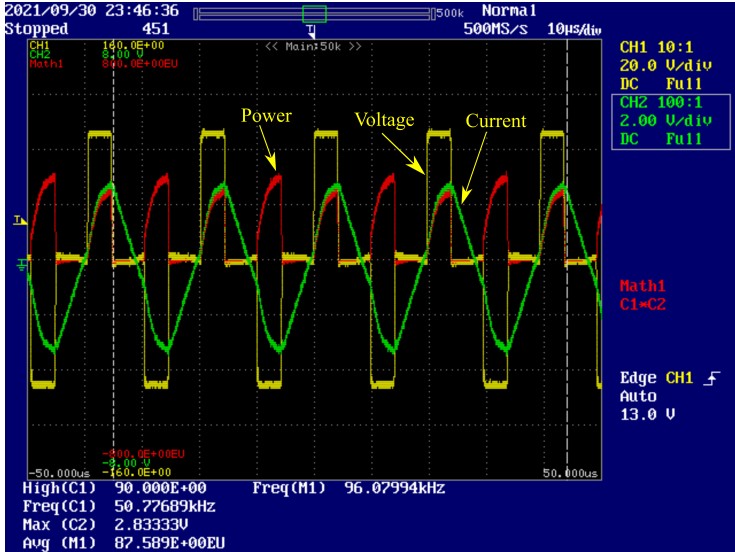

**Figure 20.** Illustration of converter output at 2 kV output voltage.

## 7. Conclusions

Parallel resonant converters (PRCs) have acquired popularity and applications in a variety of fields due to their excellent performance. PRCs have been advantageous for years as a result of the utilization of transformer parasitics. They are not, however, challenge-preserved. The intricacy of the models, as well as the challenges of implementing them in small micro-controllers, are among the drawbacks. The goal is to simplify the model such that it may be implemented in a small digital signal controller. This work illustrates the static model graphic representation of PRC. The same model was decomplexed by developing a polynomial model. The last is achieved by applying data-fitting and B-spline methods to static model graphs. Two polynomials were obtained and provide a suitable frequency and duty at any level of output voltage and power. They run the prototype in the optimum mode on F28335 micro-controller. The model was simulated in Simulink and practically tested on both low and high voltages; the results in both cases are shown in this work. The general polynomials were formed, and the corresponding control program

was developed in a model-based manner, using Simulink. The program fits and runs on the F28335 without any problem, and the control waveforms are generated as expected. The same technique illustrated in this paper will be used in further processes, such as implementing the model in FPGA, creating closed-loop fuzzy control, and implementing alternative control strategies, such as constant power control.

**Author Contributions:** This paper makes up part of the research carried out by the research team that will be here mentioned. The respective contributions of each team member will be mentioned as well. E.R. made conceptualized the research, mathematical models, simulations, and program conception, equationilization, prototyping, and writing. J.D.G. developed the fundemantal relationship among various dependent and independent parameters, supervised, and validated the resaerch. P.J.V.S. tested, upgraded, and validated the prototype. J.A.M.-R. proposed the polynomial concept of generating controlling frequency and duty. A.M.P. designed and developed the used transformer. All authors have read and agreed to the published version of the manuscript.

**Funding:** This research was funded by FUNDACION PARA LA INVESTIGACION CIENTIFICA Y TECNICA FICYT, grant number SV-PA-21-AYUD/2021/50938 and It was partially supported by Africa Improved Foods https://africaimprovedfoods.com, in the skills development program.

**Conflicts of Interest:** The authors declare no conflict of interest. The funders had no role in the design of the study; in the collection, analysis, or interpretation of data; in the writing of the manuscript, or in the decision to publish the results.

## Abbreviations

The following abbreviations are used in this manuscript:

| | |
|---|---|
| CCS | Code Composer Studio |
| IGBT | Insulated Gate Bipolar Transistor |
| PRC | Parallel Resonant Converters |
| PWM | Pulse Width Modulation |
| TI | Texas Instrument |

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
