# Peer review of "A Parallel Resonant Converter Polynomial Model Implemented in a Digital Signal Controller"

_electronics, doi:10.3390/electronics11071085_

Round 1

Reviewer 1 Report

It seems to me:

  1. The model, and control are feed-forward.
  2.  Therefore, it does  not provide a suitable control regarding changes in the load, or in the dc input voltage, e.g., battery charger.
  3. As a result, the manuscript worth is rather limited. 

Author Response

Thank you very much for your review, observation comments and suggestions. They ake a major contribution to the improvement of this work. I appreciate.

Here below are the responses and clarifications to your observations.

  1. The model, and control are feed-forward.

Ans. Thank you very much for the observation. The topology is still open-loop and the control strategy has not been proposed yet (this is planned in the future improvements of this project). It could provide, for instance, a start-up strategy (quite common in X-Ray, medical applications) guiding the power supply to a known set point, or it could generate alarms if the currents or voltages measured are larger than expected. Just now, being the input voltage the unique  input, it is only possible to implement a feed-forward, like in any other converter. If the output voltage, and/or resonant current, output load, etc are inputs to the digital controller, it is possible to use different control strategies. For instance, if the output voltage is measured, it is easy to implement a constant-voltage power source, following “optimum paths” for instance.

2. Therefore, it does  not provide a suitable control regarding changes in the load, or in the dc input voltage, e.g., battery charger.

Ans. This has been answered before, please consider the answer in comment no.1.

3. As a result, the manuscript's worth is rather limited. 

Ans.

The paper proposes a valid methodology to implement the control using digital controllers (DSCs, FPGAs, etc):

-          Based on a given model, a polynomials set is obtained

-          MathLab model based on the previous set

-          Upload directly to DSC, FPGA, etc 

One of the most important advantages is that the model is based on normalized parameters so it is only necessary to input resonant frequency: it is not necessary to modify the firmware for every converter.

The fuzzy closed control is more suitable for the proposed model. It is always possible to follow the optimum mode in any change in output voltage. It is a completely open system that allows the implementation of whatever possible digital control.

Reviewer 2 Report

In this paper, a  parallel resonant converter polynomial model implemented in a digital signal controller is proposed.

The paper contains scientific insight, anyway, some parts are not clear and some improvements are required:

1. It is suggested to remove the acronyms (PRC) and (DSC) from the title.

2. Line 12: "...widening of the operating span. And ensures the requisite...". Please remove the ".And" and merge the sentences.

3. Figures should be in order. The first Figure mentioned in the text is Fig. 2 (line 31).

4. Line 40: it is not clear the advantages of the proposed technique with respect to [16]. The novelty should be highlighted.

5. Is Ls the primary leakage inductance while Cp the parasitic resistances? Please explain it. In addition, does the magnetizing inductance has been taken into account?

6. Why the secondary leakage inductance is not taken into account? Please justify this.

7. From Fig. 2 looks like the inverter operates in soft-switching conditions only during turn-on (ZCS). Please explain Fig. 2.

8. More information related to the equations used to extrapolate the curves shown in Fig. 4 and 5 should be provided.

9. Since the power converter operates at a high switching frequency, the parasitic components of the transformer can be exploited to reach soft switching. Anyway, in Section I ( Grapical Representation of PRC Static Model), it is not mentioned the model of the power switch which has been used to extract the curves shown in Fig. 4. Does it has been used as an ideal switch? If yes, it should be specified highlighting that more complex models can be easily integrated as future steps:

- B. Nguyen, X. Zhang, A. Ferencz, T. Takken, R. Senger and P. Coteus, "Analytic model for power MOSFET turn-off switching loss under the effect of significant current diversion at fast switching events," 2018 IEEE Applied Power Electronics Conference and Exposition (APEC), 2018, pp. 287-291, doi: 10.1109/APEC.2018.8341024.
- E. Locorotondo et al., "Analytical Model of Power MOSFET Switching Losses due to Parasitic Components," 2019 IEEE 5th International forum on Research and Technology for Society and Industry (RTSI), 2019, pp. 331-336, doi: 10.1109/RTSI.2019.8895562.
- L. Dobrescu, R. Smeu and D. Dobrescu, "Load switch power MOSFET SPICE model," 2016 International Conference and Exposition on Electrical and Power Engineering (EPE), 2016, pp. 644-647, doi: 10.1109/ICEPE.2016.7781418.

10. It is suggested to use the grid in all the plots.

11. In Table 4, the same number of significant digits should be used for all the numbers.

12. Which can be future improvements? Please include them in the conclusion section.

Author Response

Thank you very much for your review, observation comments and suggestions. They make a major contribution to the improvement of this work. I appreciate.

Here below are the responses and clarifications to your observations.

  1. It is suggested to remove the acronyms (PRC) and (DSC) from the title.

Ans. The Title was changed from A Parallel Resonant Converter (PRC) polynomial model implemented in a Digital Signal Controller (DSC)” to “A Parallel Resonant Converter polynomial model implemented in a Digital Signal Controller”. The acronyms were removed as suggested.

  1. Line 12: "...widening of the operating span. And ensures the requisite...". Please remove the ".And" and merge the sentences.

Ans. The Line 12 “.And” and the sentence merging resulted in the following: “...widen the operational range and generalize the model. It also offers the essential protection…” as shown on Line 10 and Line 12.

  1.  Figures should be in order. The first Figure mentioned in the text is Fig. 2 (line 31).

       Ans. The order of figures is modified as suggested.

  1. Line 40: it is not clear the advantages of the proposed technique with respect to [16]. The novelty should be highlighted.

Ans.  The clarification on the advantages of the proposed technique with respect to [16] was highlighted in the paper, please refer to “diff.tex” or “diff.pdf” for tracking the modifications.

  1.  Is Ls the primary leakage inductance while Cp the parasitic resistances? Please explain it. In addition, does the magnetizing inductance has been taken into account?

AnsLs is the leakage inductance (we consider it at the primary side)

Cp is the parasitic parallel capacitor

The transformer model is the typical for High-Voltage applications, based in a serial inductor and a parallel capacitor; as long as the secondary winding is usually large, a capacitor appears at the secondary side, and it is reflected to the primary side.

Due to the relative values, magnetizing inductance is considered as infinite; in other words, it does not contribute to the resonance, for this reason, is not considered

  1. Why the secondary leakage inductance is not taken into account? Please justify this.

Ans. It is considered in the primary, in such a way that it is possible to concentrate both of them in only one

  1. From Fig. 2 looks like the inverter operates in soft-switching conditions only during turn-on (ZCS). Please explain Fig. 2.

AnsThe converter behavior is explained in this paper:

Díaz González, Juan; Villegas Saiz, Pedro José; Martín Ramos, Juan Antonio; Martín Pernía, Alberto; Martínez Esteban, Juan Ángel. "A high-voltage AC/DC resonant converter based on PRC with single capacitor as an output filter". IEEE Transactions on Industry applications 46(6). pp. 2134 - 2142. Institute of Electrical and Electronics Engineers, 2010. ISSN 0093-9994.

It is the reference number 11. As long as the operating modes have been already described, we didn't include them here.

  1.  More information related to the equations used to extrapolate the curves shown in Fig. 4 and 5 should be provided.

Ans.. The equations used to extrapolate the graphs in Figures 4 and 5 are detailed in the following paper:

Díaz González, Juan; Villegas Saiz, Pedro José; Martín Ramos, Juan Antonio; Martín Pernía, Alberto; Martínez Esteban, Juan Ángel. "A high-voltage AC/DC resonant converter based on PRC with single capacitor as an output filter". IEEE Transactions on Industry applications 46(6). pp. 2134 - 2142. Institute of Electrical and Electronics Engineers, 2010. ISSN 0093-9994.

It is reference number 11.  And more information is shown in the paper, please refer to “diff.tex” or “diff.pdf”, to track the modification.

  1. Since the power converter operates at a high switching frequency, the parasitic components of the transformer can be exploited to reach soft switching. Anyway, in Section I ( Graphical Representation of PRC Static Model), it is not mentioned the model of the power switch which has been used to extract the curves shown in Fig. 4. Does it has been used as an ideal switch? If yes, it should be specified highlighting that more complex models can be easily integrated as future steps:

- B. Nguyen, X. Zhang, A. Ferencz, T. Takken, R. Senger and P. Coteus, "Analytic model for power MOSFET turn-off switching loss under the effect of significant current diversion at fast switching events," 2018 IEEE Applied Power Electronics Conference and Exposition (APEC), 2018, pp. 287-291, doi: 10.1109/APEC.2018.8341024.

- E. Locorotondo et al., "Analytical Model of Power MOSFET Switching Losses due to Parasitic Components," 2019 IEEE 5th International forum on Research and Technology for Society and Industry (RTSI), 2019, pp. 331-336, doi: 10.1109/RTSI.2019.8895562.

- L. Dobrescu, R. Smeu and D. Dobrescu, "Load switch power MOSFET SPICE model," 2016 International Conference and Exposition on Electrical and Power Engineering (EPE), 2016, pp. 644-647, doi: 10.1109/ICEPE.2016.7781418.

Ans. In the analysis, ideal switches have been considered; this assumption does not affect the final results so much. It is also true that affects in terms of efficiency, EMI, etc, and it is sure that a more accurate model can be obtained if the model switches are included. Your suggestion is included in the paper, please refer to “diff.tex” or “diff.pdf” for tracking the modifications.  Thank you very much.

  1.  It is suggested to use the grid in all the plots.

Ans.  Grids were added on plots in Figure 5, Figure 6, Figure 7 and Figure 8

  1. In Table 4, the same number of significant digits should be used for all the numbers.

Ans. The decimals were fixed to 4 digits in Table 3 and Table 4. Please refer to “diff.tex” or “diff.pdf” to track the modifications in the mentioned tables

  1. Which can be future improvements? Please include them in the conclusion section.

Ans. 

The following are the future improvements and future works:

  • Implementing the model in a FPGA
  • Implementing closed- loop Fuzzy control of Parallel Resonant Converter
  • Implementing different control strategies: Constant power

They are included in the paper : Please refer to “diff.tex” or “diff.pdf” in the conclusion part to track the modififications

Reviewer 3 Report

This paper proposes a  polynomial model for a parallel resonant converter and its implementation  in a digital signal controller.

The proposed methodology seems correct, the experimental prototype and its behaviour is interesting.

However the authors must show how this work contribute to the advancement of the state of the art with comparisons and related discussions.

English must be accurately checked and the Abstract and Conclusion sections must be better organized.

Author Response

Thank you very much for your review, observation comments and suggestions. They make a major contribution to the improvement of this work. I appreciate.

Here below are the responses and clarifications to your observations.

Comments and Suggestions for Authors:

This paper proposes a polynomial model for a parallel resonant converter and its implementation in a digital signal controller.

The proposed methodology seems correct, the experimental prototype and its behaviour is interesting.

Thank you very for your comment, it is appreciated

  1. However the authors must show how this work contribute to the advancement of the state of the art with comparisons and related discussions.

Ans In [16] the starting model is an approach; in our case, the starting model is the real one. The whole process is shown in this paper, how to obtain a normalized model in a DSC. Since the model is obtained using normalized parameters, it is only necessary to input the resonant frequency and the input voltage to customize the controller, it is not necessary to upload a new model every time the converter parameters change. Once the model is implemented, it opens a wide range of control possibilities: in the DSC an analog A/D is available, so it is possible to read the output current, voltage and implement different control strategies. 

2. English must be accurately checked and the Abstract and Conclusion sections must be better organized.

Ans. English was checked and  abstract and conclusion were re-orgaized. Some modifications were done on indicated the  points: Please refer to the “diff.tex” or “diff.pdf” to see the modifications

Round 2

Reviewer 1 Report

The authors' response did not answer the reviewer concerns.

Reviewer 2 Report

All the requested changes have been performed. The quality of the paper has been significantly improved.

Reviewer 3 Report

The authors have done their best to respond to my suggestions and comments and have done it enough.

Basically this manuscript could be published in its present form.